# Ownership, Governance, Uses, and Ecosystem Services of Community Forests in the Eastern United States

Meredith Hovis [1,*], Gregory Frey [2], Kathleen McGinley [3], Frederick Cubbage [4], Xue Han [5] and Megan Lupek [4]

1 Udall Center for Studies in Public Policy, University of Arizona, Tucson, AZ 85719, USA
2 USDA Forest Service, Southern Research Station, Research Triangle Park, NC 27709, USA
3 USDA Forest Service, International Institute of Tropical Forestry, Río Piedras, PR 00926, USA
4 Department of Forestry and Environmental Resources, North Carolina State University, Raleigh, NC 27695, USA
5 Department of Management, Harbin Finance University, Harbin 150028, China
* Correspondence: hovis@arizona.edu

**Abstract:** Over time, community forests (CFs) have been established across the globe to meet various social, economic, and ecological needs. Benefits of CFs include conserving resilient forests and natural resources and ecosystem services, enhancing social and economic capital, and leveraging local and indigenous knowledge in forest and natural resource management and decision-making. Research on CFs in the U.S. is quite limited, and cases that have been assessed show a wide spectrum in terms of CF ownership, organizational structure, governance, property rights, and uses. Through an exploratory research approach, this study enhances the understanding of the characteristics of CFs in the U.S. and the ecosystem services and other benefits that they provide. Through online web searches, we compiled one of the first comprehensive lists of CFs in the Eastern U.S. Prior to this study, there was no publicly available comprehensive database or list of CFs in the country. Subsequently, we conducted comparative case study research, which included semi-structured in-person interviews with key stakeholders from four CFs in the Eastern U.S. to understand CF ownership, governance, uses, and benefits. CFs benefits frequently included cultural services, such as recreation and education, and regulating and supporting services, such as water quality and wildlife habitat. Much less common was a focus on provisioning services such as timber or non-timber forest products. Maintaining collaboration and funding for CF efforts in the long run without significant CF revenues remains a challenge for most forests. Overall, this research sheds lights on CF characteristics and capacities in the Eastern U.S. and identifies potential opportunities and needs for the U.S. in the future. CFs researchers, managers, and community members.

**Keywords:** community forests; Eastern United States; recreation; ecosystem services; governance; public participation

## 1. Introduction

In 2017, forests covered 766 million acres (310 ha) in the United States [1]. Although the population has increased in the U.S. substantially since 1910, the total forest land area has been relatively stable. The US forest ownership consists of 58% private forest lands (i.e., 34% private noncorporate or family forest lands, 20% corporate forest lands, 2% tribal forest lands, and 2% other private forest lands) and 42% public forest lands (i.e., 31% federal forest lands; 9% state, and 2% local government forests) [1]. The perspectives of public and private owners are too diverse to characterize narrowly, but one might surmise that industrial forest owners focus the most on commodity production such as timber, family forest lands use lands for commodities, investments, recreations, and amenities. Public lands in the U.S. may focus more on ecosystem services and recreation but retain some timber and commodity production, especially on state lands.

Despite a relatively stable land area, forests are faced with an expanding and intersecting host of issues, such as drought, wildfire, floods, and invasive species, all of which are intensified by global climate change, as well as deforestation from land-use changes and alterations [2,3]. Traditional forest policy responses to forest threats have included public ownership, regulation, education, and incentives. These focused mainly on timber and non-timber products in the past, but broader sets of ecosystem services provided by forests have become increasingly important. For example, expanding Federal government programs (e.g., Forest Legacy Program, Forest Stewardship Program) pay private forest landowners to encourage forest retention and sustainable management practices to enhance and restore ecosystem services [3].

Government ownership and management of forests and natural resources were initially implemented to protect natural resources from overexploitation. Private markets have yet to provide the socially optimal level of ecosystem goods and services from forests since there are few active markets and prices for the broad range of these goods and services. In response, various private, nongovernment, and government organizations have developed programs that provide payments for ecosystem services from forests and other natural resource systems, and voluntary environmental certification programs intended to incentivize sound resource management and protection through market access, product labeling, and price differentials.

Forests provide a large number of market commodities in the US, primarily through timber products and developed recreation, as well as a vast array of non-market commodities such as air, water, carbon storage, and biodiversity. This mix of forest goods and services requires both private markets and community management [4]. New forms of natural resources governance and management have evolved to address expanding knowledge and values of forest goods and services [5,6]. Among these new approaches are community forests (CFs), involving a broader role for local communities in forest decision-making and benefits. CFs have been put forth as an innovative prospect for enhancing forest conservation and local livelihoods in ways that traditional policy tools do not typically provide.

CFs took root in the international context in the late 1970s to meet the needs of rural populations [7,8]. In 1978, the UN's Food and Agriculture Organization (FAO) described "community forests" as a means for "provision of fuel and other goods essential to meeting the basic needs at the rural household and community level, provision of food and environmental stability necessary for continued food production, and the generation of income and employment in rural communities [9]".

Although the development and study of CFs have expanded globally since the 1970s, there is no universal definition of "community forest", because of differences in policy, history, culture, ecology, and other contexts [10–12]. In the U.S., tribal lands and New England town forests represent historical models of CFs, with more recent models including CF across a range of public and private ownerships [8,13–15]. Generally, researchers and scholars agree that CFs are defined by the following characteristics: (1) local populations have a substantive role in decision-making, (2) local values are incorporated into management and governance, and (3) CFs are managed to deliver market and nonmarket benefits to local people [11,16–19]. In the U.S. context, what makes CFs different from private or government-owned forestlands is the role that local community members play in forest decisions and outcomes, such as determining the objectives and purpose of the CF, managing and monitoring the property, and receiving social and economic benefits [8,20].

There is relatively little literature that compares the characteristics, status, and outcomes of CFs in the U.S., and examples across the U.S. portray a diverse spectrum of CF forms [15,17,21,22]. The refereed literature on CFs consists mainly of studies from the global South where marketable benefits are the community's primary focus (e.g., [23,24]). Some literature has been recently published on CFs in the Western U.S. (e.g., [25,26]), and others have focused on U.S. New England town forests (e.g., [8]).

CFs encompass organizational and governance mechanisms that may protect and provide a broad range of ecosystem services at multiple scales. The Millennium Ecosystem Assessment (MEA) (2005) developed the standard approach for classifying ecosystem services [4]. Supporting services include nutrient cycling, soil formation, and primary production. Provisioning services include commodities such as food, fresh water, wood and fiber, and fuel. Regulating services include climate regulation, flood regulation, disease regulation, and water purification. Cultural services include aesthetic, spiritual, educational, and recreational [4]. These various services are linked in multiple ways to well-being—security, the basic material for a good life, health, good social relations, and freedom of choice and action [27]. We utilized the MEA approach to ecosystem services to frame our research questions, methods, and instruments.

The purpose of this study was to explore the diversity of self-identified "community forests" in the Eastern U.S. (including Puerto Rico) in terms of: ownership, organizational structure, governance, property rights, and forest uses (i.e., ecosystem services). To better understand these topics, we explored the background, history of the establishment, current practices and activities, and community perceptions of a subset of CFs in the Eastern U.S. The results shed light on the links between these key characteristics of CFs and ecosystem services objectives and outcomes. In addition, the study results can help CF managers, local volunteers, managers, practitioners, and researchers in the U.S., and potentially across the globe, to understand the best establishment and management practices of CFs and how they benefit local communities.

## 2. Methods

We conducted exploratory social science research to address our research objective. First, we collected data on key characteristics of CFs in the Eastern U.S. and the range of ecosystem services that they aim to provide and protect from self-identified "community forests" from primary and secondary sources available on the internet. Second, we conducted in-depth comparative case study research to gain detailed information from four selected CFs representing a range of CF characteristics and outputs.

### 2.1. Compilation of Online Information

Through online web-based searches, we assembled a list of CFs in the Eastern U.S. (east of the Great Plains and including Puerto Rico). We selected the Eastern U.S. due to the lack of current refereed literature on CFs in the region, except for some research on town forests in the U.S. Northeast (e.g., [8]).

The online search provided publicly available information about the CFs and formed a basis for selecting comparative case studies for the second phase of this research. At the time of initial data collection, there was no publicly available comprehensive database or list of CFs in the country. However, several organizations had compiled shorter lists of CFs, typically where they had assisted in land acquisition or establishment of governing institutions or conducted other preliminary/exploratory work (e.g., Trust for Public Lands, U.S. Endowment for Forestry and Communities, the Ford Foundation, USDA Forest Service [28–36]). These documents and lists formed the basis for our iterative web search. As the search proceeded, other initiatives and forests came to light, which were added.

We conducted a systematic search of publicly available and accessible websites and documents. Systematic searches are common in social science research and are utilized increasingly to initiate qualitative research [37–39]. We identified key categories of variables about forest ownership, organizational structure, governance, management, uses, and benefits from the forestry, CF, ecosystem services, and nonprofit management literature (e.g., [32–36]) that we used to guide our online search and generally characterize CFs. We used keywords plus each CF's name to discover further information about each forest. Examples of the keywords included "management", "ownership", and "uses", We included an example of our keyword search in Appendix A. We reviewed each CF webpage (if available online) and documents, such as management plans, meeting minutes, new reports, and annual reports.

## 2.2. Comparative Case Studies

We then selected four CFs to conduct in-depth case studies involving site visits, field observations, in-person semi-structured interviews with key CF stakeholders, and archival document collection and review. The CFs selected included Río Hondo in Mayagüez, Puerto Rico; Nine Times in Pickens, South Carolina; Birch Ridge in New Durham, New Hampshire; and Page Pond in Meredith, New Hampshire. All are located in the Eastern U.S. and received funding from the U.S. Forest Service Community Forest Program (CFP). This descriptive work helps identify some prevalent CF characteristics and capacities in the Eastern U.S. This investigation of this newer category of U.S. forest ownership and management described some key elements of CFS, and identifies opportunities for future CF researchers, managers, and community members.

We selected the four CFs from our list to learn more about the interactions between CF ownership, governance, management, and ecosystem services objectives and outcomes through a comparative case study approach. The four were chosen to be both representative and diverse, thereby highlighting variability and diversity among CFs in general, and providing rich, contextual data to give insight into the CF background, history of the establishment, current practices, and community perceptions of a subset in the Eastern U.S. The case studies were diverse in terms of their ownership status, geographic location, forest uses, forest size, and management objectives from what we discovered via online searches. We selected the cases purposely to represent a range in scale, geography, tenure arrangements, and socioeconomic benefits. A multiple case study allows for meaningful comparison and exploration of differences and diversity [40–42].

Each case study employed three types of qualitative data collection modes: (1) semi-structured interviews with key informants, (2) document review of official records, financial reports, and other relevant documents to the organization, and (3) systematic web-search of each CF project's online documents and website. This approach triangulates data sources to increase the validity of findings [43–45]. From June to August 2019, we traveled to the communities, spending approximately one week in each location.

We drew from the theory and empirical evidence, themes, and concepts presented in the CF literature to develop a semi-structured interview to elicit respondents' experiences and points of view on CFs (Interview Questionnaire shown in Appendix B). We asked various questions about CF uses and ecosystem services, ownership status, governance and management activities, public input strategies, and strengths and challenges.

We interviewed volunteers, landowners, managers, community members who live near the forest, and local elected officials who handle CF duties. This mix of managers to community members to users is a standard approach in qualitative research, in order to gain different perspectives from stakeholders about the operation, merits, outcomes, and effects of a program or system [45]. We interviewed a total of 18 participants from the case study locations; eight from Río Hondo in Mayagüez, PR, USA; one from Nine Times in Pickens, SC, USA; six from Birch Ridge in New Durham, NH, USA; and three from Page Pond in Meredith, NH, USA.

We obtained a list of documents from each case study to further inform its context in terms of land ownership and protection status; history and development; governance and associated institutional structures; finances, funding, and capital assets; collaboration and partnerships; management practices; outputs, outcomes, and benefits distribution.

We developed a common coding guide, initially consisting of predetermined codes based on the CF literature, study objectives, and case study questionnaires in a deductive approach to the data analysis. As we proceeded through the analysis and coding of case study interviews and other documents, we developed additional codes inductively based on emergent topics and variables discovered from the data through the analytical process [46,47]. We organized the coding guide by six major themes and continued developing and refining the codes from the data until no new or revised codes occurred or emerged (Figure 1). Once each case study was reviewed, we compared the overall themes across the cases.

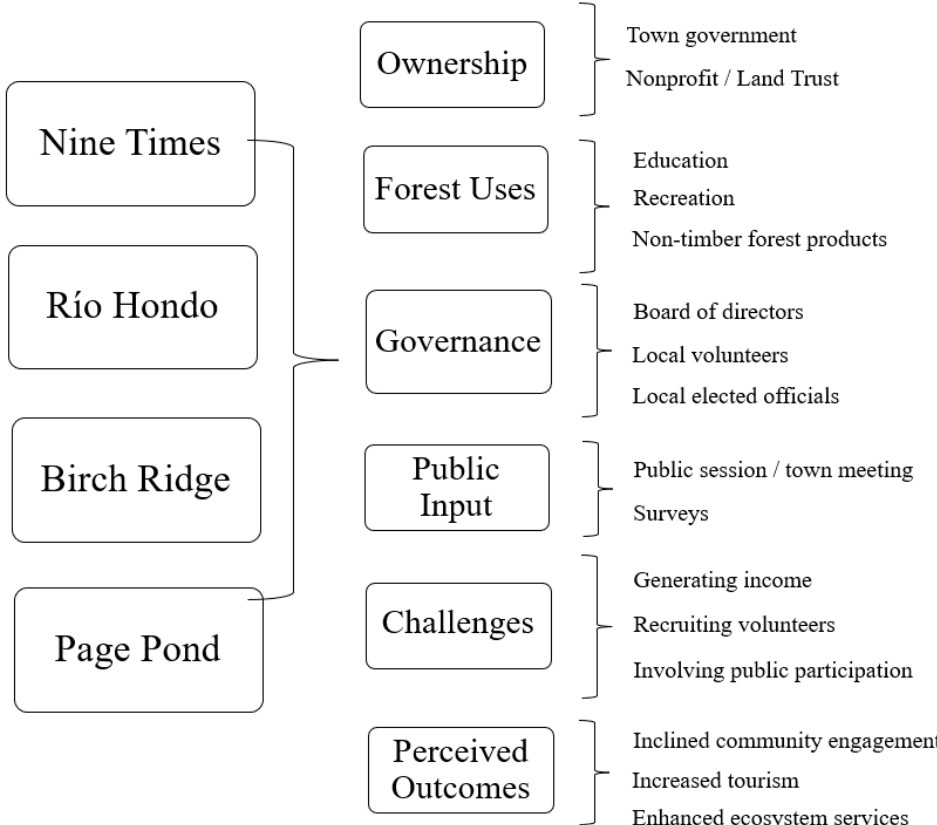

**Figure 1.** Qualitative data coding tree.

### 3. Results

*3.1. Descriptive Statistics*

We discovered 70 CFs in the Eastern U.S. through online searches (Appendix C). Descriptive statistics are indicative but not conclusive about CFs in the Eastern U.S. We found that 47% of the projects made their management plan available online, and 8% had bylaws accessible to the public. Fifty percent referred to working with the public or community in on-the-ground management or decision-making processes, and 80% referred to collaborating with other organizations, such as nonprofit organizations, land trusts, private entities, local businesses, or governmental agencies.

The median area of the eastern CFs was approximately 142 hectares (ha) (350 ac). The first established CF in the East was the Menominee Forest in Neopit, WI, USA, in 1854. The median year of establishment of the Eastern U.S. CFs was 2011. A number of CFs were established decades ago, particularly in the northeast, and many more CFs have been established since the 2008 Farm Bill (Food, Conservation, and Energy Act of 2008; Publ. Law 110–234), which created the U.S. Forest Service's Community Forest and Open Space Conservation Program (also referred to as the Community Forest Program (23)) [31]. The CFP grants tribal entities, local governments, and nonprofit organizations funding to acquire the land to become a CF. Sixty-three percent of eastern CFs identified through the search received funding from the CFP, with the mean amount received from CFP of USD 202,235.

CFs can be owned by various public or private entities [15,48,49]. Most (*n* = 35) of the eastern CFs identified are owned by local governments, and many (*n* = 25) of the remaining eastern CFs are owned by nonprofit organizations, such as land trusts. Seven are jointly owned, with a combination of town government and nonprofit organizations. Two of the eastern CFs in our database are owned by tribal entities. Only one is owned by the state or federal government, and none of the eastern CFs are owned by a private for-profit entity.

The majority of CF websites list that their purposes include forest access for cultural ecosystem services, including recreational purposes (93%) such as hiking, biking, snowmobiling, or horseback riding, as well as education (69%). Many projects (87%) reference on their website the importance of regulating and supporting ecosystem services that the CF provides for their community, such as clean water and air, carbon sequestration, and vital biodiversity habitat. Many of the CF's websites also reference forest provisioning ecosystem services, including harvesting and generation timber forest products (64%) and some for non-timber forest products (21%).

### 3.2. Qualitative Comparative Case Studies

Generally, there are three purposes for establishing CFs in the U.S.: (1) to secure local access to various goods and services, (2) to allow for public participation and input in decision-making and management processes, and (3) to protect the property from conversion to non-forest uses [8,36,50–52]. Based on the list of CFs in the eastern U.S. and the extensive online literature search, we selected our case studies that illustrate the application of these three principles to varying degrees.

The four case studies represent a range of ownership types, management objectives, and governance regimes (Table 1). We identified interview participants through managers and key players from each CF. For anonymity, interviewees are represented in the text with an acronym (RH = Rio Hondo, NT = Nine Times, BR = Birch Ridge, PP = Page Pond) and a number.

**Table 1.** Community Forest Case Study Sites Overview.

| Community Forest | Río Hondo, Mayagüez, PR, USA | Nine Times, Pickens, SC, USA. | Birch Ridge, New Durham, NH, USA | Page Pond, Meredith, NH, USA |
|---|---|---|---|---|
| **Characteristics** | | | | |
| Size (Hectares) | 28 | 667 | 815 | 310 |
| Town Population | 73,336 | 3169 | 2693 | 1495 |
| Forest Type | Secondary subtropical; urban forest | Mixed hardwood and pine | Birch and northern hardwoods | Wetland; upland forest |
| Year Established | 2018 | 2012 | 2019 | 2009 |
| Ownership | Local entity—Town of Mayagüez | Nonprofit entity—Naturaland Trust | Nonprofit entity—Southeast Land Trust | Local entity—Town of Meredith |
| Administration | Río Hondo CF Board of Directors, Town of Mayagüez | Naturaland Trust Board of Directors | Southeast Land Trust | Town of Meredith Conservation Commission |
| Forest Manager | Río Hondo CF Board of Directors, Town of Mayagüez, local community members | Naturaland Trust Board of Directors | Southeast Land Trust, Moose Mountain Regional Greenways, Powder Mill Snowmobile Club, local community members | Town of Meredith Stewards |
| Forest Use | Ecological services, education, recreation, NTFPs | Recreation, timber | Ecological services, education, recreation, timber | Ecological services, education, recreation, timber |

Notes: NFTPs abbreviated for non-timber forest products; Population reported from 2020 Census.

### 3.2.1. Bosque Comunitario de Río Hondo (Río Hondo Community Forest)

The initial vision of the Río Hondo effort was to create an eco-agritourism site in Mayagüez, Puerto Rico, in 2005. A local champion, RH4, organized the effort and recruited local individuals to join. The original idea was to create a destination for tourism with a café, a demonstration site for tobacco and sugar cane farming, a greenhouse for native

plants and herbs, and a protected forested area on the property that could attract tourists and enhance the town's social and economic prosperity, and protect the forested area. At the time, a family-owned the mostly forested 28-ha (68-ac) property.

In 2005, RH4 and the recruited members promoted the idea of an eco-agritourism site to the rest of the town and created a non-profit organization, called Proyecto Comunitario Agro-Eco-Turístico Barrio Río Hondo, Inc. The organization consisted of a president and board members. With support from the local community, they pitched the idea to the municipality, who supported the idea but did not have sufficient funds to purchase the land, which was listed at USD 800,000.

In 2007, with help from professors at local universities, the organization developed a business and management plan for the site to be used for fundraising and finding investment partners. Eventually, the property owners granted a leasing contract to Proyecto Comunitario Agro-Eco-Turístico Barrio Río Hondo, Inc. (Río Hondo, Mayagüez, PR, USA) for the 10% non-forested portion of the property to provide access to the site while acquisition funds were secured. The organization installed a trailer on site for community meetings and workshops and established three greenhouses where plants and herbs were grown and sold for fundraising. For nearly a decade, the organization knocked on doors in the local community and across the island to raise money for the acquisition and eco-agritourism site without success.

Eventually, the organization learned of the USFS CFP and submitted a proposal for funding their project in 2014. Given the objectives and requirements of the CFP, development aspects of the proposal (e.g., agricultural development, café) had to be eliminated and an increased focus on sustainable forestry and natural resource management aspects had to be incorporated in the proposal to be considered for funding. Ultimately, Proyecto Comunitario Agro-Eco-Turístico Barrio Río Hondo, Inc. was granted USD 400,000 from the CFP for forest acquisition, and the municipality agreed to pay the remaining USD 400,000 to purchase the land. RH7, a member and volunteer of the Río Hondo CF recalled:

> "It was difficult to change the plans, but we did not have enough money to buy the forest without the grant's help. It was too much money for a nonprofit. The main goal was always to conserve the land. We just had to change the plan based on their restrictions".

Río Hondo is the first CF established with CFP support, owned by the municipal government, and managed by a local NGO in Puerto Rico. The property is surrounded by suburban and rural subdivisions and is approximately a five-minute drive to downtown Mayagüez. At the time of data collection, a small number of community members work daily at the property as unpaid volunteers. The 90% forested portion of the property is used for cultural services, such as recreation and education. Students from the University of Puerto Rico and Consuelo Pérez Cintrón Elementary School visit the property for research and academic purposes. RH5, a board member, states, "it is important to connect the children in the community with nature. The students can work in the forest for school credits or volunteer hours." Another volunteer, RH2, is one of the leading tour guides of the forest and notes that they "have approximately four school group visits each month". The Río Hondo Community Forest has also hosted AmeriCorps Program participants and Boy Scout groups; both helped develop trails in the forest. The CF sells trees, herbs, vegetables, other plants from the greenhouse, and sofrito (a Puerto Rican recipe made from herbs) to raise additional income.

The CF board of directors meets once a month, volunteers make and sell sofrito for additional funding, and local school groups and university students visit for educational and recreational events. The CF lands are managed by a board of director members and volunteers from the neighborhood.

Hurricane Maria struck Puerto Rico on 20 September 2017, resulting in widespread damage throughout the region. It has been cited as the costliest disaster in Puerto Rico's history and the third costliest in U.S. history [53]. Hurricane damages at the Río Hondo Community Forest included uprooted, broken, and defoliated trees throughout the forest,

total destruction of the greenhouses and total loss of its produce, and damaged and destroyed trail signage and other infrastructure throughout the property.

As described by RH5:

"We do what we can since Maria and the destruction that occurred. It has been a setback. But, this year, we will contact and reach out to more students from the university and nearby high schools and elementary schools. We plan to work on more grant proposals and complete the forest management plan".

3.2.2. Nine Times Community Forest

Nine Times Community Forest, also referred to as Nine Times Preserve per the entrance signs, is one of eight properties owned by Naturaland Trust. From the 1920s to 2012, the 667-ha (1648-ac) property was owned and logged by a local timber company. In 2007, the timber company put the property on the market. Naturaland Trust and partnering organizations were interested in purchasing the property, but they were half a million dollars short of the purchase price. Eventually, in 2012, the company dropped the asking price to about half of its 2007 listing price, making it more accessible for purchase by the organization.

In 2012, Naturaland presented the project to the CFP for acquisition funding. The proposal was approved, and these funds were leveraged to raise an additional USD 3.1 million to acquire the property later that year. Duke Energy, an electric power holding company, also holds an easement on the property. Duke's powerlines run through the middle of the forest, and they own the rights to clearing out pathways for their vehicles and spraying herbicides under the powerlines to prevent growth.

A local community member, NT1, is responsible for managing the property. As an unpaid volunteer, NT1 maintains all trails and signage and visits the CF about three times a week. Occasionally, the manager recruits individuals and local user groups to help with trail maintenance. NT1 was essentially a one-person manager and caretaker, assisted by ad hoc efforts mostly by recreation user groups.

The forested areas on the property consist of approximately 23% planted pine, 9% natural hardwood, and 68% mixed hardwood and pine forest. Since Naturaland's ownership of the property began, the organization has hired McMillian Logging Company to harvest some of the forest's timber, providing income for the CF.

The CF provides mainly cultural services, such as hiking, hunting, and rock climbing. They receive approximately USD 7000 to USD 8000 per year from Wildlife Management Areas (WMA) through hunting leases on the property. Naturaland estimates that there are 3500 visitors per year, consisting of approximately 1500 rock climbers, 1000 hunters, and 1000 hikers. The Climbing Coalition attracts the most people to the area. NT1 states, "There are a lot of social media events, mainly through Facebook, to recognize the rock-climbing club, and there are a lot of hiking clubs and online forums that visitors will post to. The word usually gets out." The property is also used for cultural services such as education from local schools, most frequently by Clemson University students. When asked about the next steps for the community forest, NT1 stated:

"We need to attract visitors and recruit someone to help bring in volunteers. Someone needs to come and take over and organize volunteers who will help with the property. I also need to put more signs and kiosks up and make a couple more parking lots to increase visibility. We'd also like to expand the property to the north and southwest to make a wildlife corridor. We hope Naturaland will buy those properties. There is also nearby private property in Pickens that we'd like to acquire".

Despite the wealth of supporting, regulating, and cultural services provided to the local community, Nine Times had the lowest levels of community participation in forest governance and management components. During our field visits, we were unable to find anyone on adjacent properties to the forest or in the local stores who were aware that Nine

Times is a CF. There was no active board focused exclusively on the CF, but rather a loose connection to the larger organizational structure of Naturaland Trust, which had various widely dispersed properties in the region.

### 3.2.3. Birch Ridge Community Forest

Birch Ridge is an 820 ha (2027-acre) forest that sits on top of a hill, overlooking Merrymeeting Lake in New Durham, NH, USA. New Durham is a town that has a small population that benefits from the seasonal influx of visitors and part-time residents, by some accounts tripling in population during the summer. Birch Ridge is owned by Southeast Land Trust of New Hampshire (SELT), which owns and manages over a dozen other properties.

Before SELT, Birch Ridge was owned by a local timber company. In 2018, there were rumors that the company would sell the property to a developer, who would convert the land to condominium housing. When BR1, a local champion and business owner, heard about the proposed sale, he approached the property manager that day. The manager supported the mission to conserve the property and agreed to sell the property for USD 2 million. Still, he needed a non-refundable deposit of USD 200,000 in the next 30 days. Confidentially, BR1 shook on the deal.

Approximately 400 customers pass through BR1's local business. BR1 was able to rally together the members of the Association, individuals who passed through the marina, small businesses, others who lived in the New Durham area, and elected town officials. BR1 states:

> "This town has VERY deep pockets . . . It was the perfect storm. This place that people care about was going to be destroyed. It was helpful that people who live on the lake have a lot of money".

In two days, the community raised USD 400,000. The next step was raising the rest to purchase the property. The town members reached out to SELT for additional help to raise the funding and to act as the primary owner. They created a steering committee, consisting of a representative from the following organizations and agencies: SELT, Town of New Durham Conservation Commission, New Durham Department of Recreation, Powder Mill Snowmobile Club, Lakes Regional Technical School, Camp Lions Pride, Farmington Fish, and Game Club, New England Mountain Bike Association, Merrymeeting Lake Association, Moose Mountains Regional Greenway, and New Hampshire Fish and Game Department. The group met monthly to plan for fundraising, community and public input events, and develop the management plan. Once the steering committee completes the management plan, they plan to elect and re-elect members and meet approximately twice a year. To assist with forming the management plan, the committee reached out to other New Hampshire CFs.

In approximately five months, SELT and the community raised USD 3 million; USD 500,000 granted through the CFP, and USD 25,000 donated from the town from the former timber owner's timber tax. BR2, a member of SELT, says that "this was the quickest turnaround on a project that we had ever seen".

The steering committee sent out surveys to New Durham taxpayers, asking questions about what recreation opportunities they wanted to see and participate in and the aspects and benefits of the CF they value most. Fifty-seven percent of the community members were most concerned with the property's impact on water quality.

In June 2019, the steering committee held its first workshop to gain the community member's input; the workshop attracted 80 attendees, a member of the Merrymeeting Lake Association, BR3, announced at the event, "This is YOUR forest!" In July 2019, the steering committee led a kick-off for the establishment of Birch Ridge Community Forest. The event consisted of guided hikes and a picnic lunch, and attracted more than 100 people. A second public input workshop was held in October 2019. Commenting on public input, BR2 explains:

"This forest is different from all of the other properties SELT owns. We need this extra step for public input before we do anything else. It can be difficult, though, because we have a lot of different cooks in the kitchen and need to hear from everyone".

Additionally, referring to the decision-making processes of the CF, BR2 says, "the steering committee discusses the community member's input from the brainstorming sessions, and then they report to the SELT board. We (referring to the SELT board) have the final say, especially regarding finances and regulations".

The property is managed by members of the Powder Mill Snowmobile Club, consisting of 400 members. In return for utilizing the property for snowmobiling, approximately 40 club members maintain 138 km of trails each month. One key player in the club, BR4, also manages game cameras on the property, which capture moose, black bears, eastern coyotes, and other wildlife. BR4 states, "the cameras help monitor the property for illegal access and usages, such as ATV and camping activities. I always catch those ATV guys". The CF is also managed by Moose Mountain Regional Greenways (MMRG), a local land trust organization in New Hampshire, which holds a conservation easement on the property.

BR5, a member of the CF committee, says, "there is a need for reforestation and regrowth due to the recent clearcutting". The volunteers and SELT members plan to create signs, kiosks and install more accessible parking lots and trails for community members and visitors. In addition, SELT hopes timber harvest can occur in the future to generate income for the sustainability of the CF.

### 3.2.4. Page Pond Community Forest

Page Pond CF started with a 229-ha (567-ac) property owned by a religious group and operated as a youth summer camp and conference facility. In 2002, the religious group planned to enhance its facilities by adding an administration building, a dining facility, camper housing, single-family and multi-family employee housing, ball fields, and waterfront improvements, including an extensive dock and a water slide into Page Pond. However, the town Department of Environmental Services did not grant the approval.

In 2008, the town negotiated a payment to purchase the property from the religious group. Meredith's town government worked closely with Trust for Public Land (TPL) to raise the necessary funds to conserve the property permanently. PP3, a TPL employee who played a large role in the acquisition process, states:

"Generally, towns do not have the experience to acquire large areas of forest land. We mainly help them fundraise and draft a management plan, but this is a really capable town, and this time we didn't have to do that much work. TPL usually has to start from scratch, but Meredith had a very engaged conservation commission and planning committee".

The town obtained enough funding through state and local grants and private donations to obtain the 229 ha—completing Phase I of Page Pond CF. PP2, an elected town official, notes, "This was the town's biggest project". Most town forest lands are donated from private properties, typically left in citizens' wills. PP2 discusses:

"The concept of a community-owned forest is not uncommon to the people in New England. Town forests have been around since the 1700s. It's very traditional to the culture. In New England, counties are the main form of government. In other parts of the country, people aren't familiar with county lands or county forests. Here, everyone has heard of a town forest".

Nine years after the acquisition of Page Pond CF Phase I, the town fought to obtain the 81-ha (199-acre) tract adjacent to the Phase I property. The Phase II expansion property encompasses approximately 61 ha of undeveloped forestland, six has of abandoned agricultural fields, and 13 ha of designated wetlands.

The CF is utilized for timber, recreation, education, wildlife habitat, and other environmental benefits. Most of the forest is a mixed lowland type, with hemlock, pine, oak,

and beech as dominants. In 2009, CF members conducted a timber harvest, generating over USD 50,000 in net proceeds. The forest was harvested extensively in 2011 and 2012, removing a large amount of marketable timber. The town hopes the Phase II parcels can be utilized for non-timber forest products (NTFP), such as a pumpkin patch or sunflower field, to attract tourism. The property also includes Page Pond, which PP3, a volunteer mentions, "as a local hotspot for fishing and swimming".

Page Pond is located less than one km from the Inter-Lakes School District (Inter-Lakes Junior and High School, Inter-Lakes Middle Tier School, and Inter-Lakes Elementary School), attracting many school groups for educational tours and field trips. Every year the MCC hosts several nature walks on the property, including educational walks with county foresters and other natural resource professionals. "We hope that we can build a sidewalk that connects the schools to the property for easy-to-access field trips or other after-school programs" PP2 notes.

The administration and governance of the CF are undertaken by the Meredith Conservation Commission (MCC), a town committee of elected officials who makes decisions about and protects the town's natural resources. The MCC consists of seven full members, five alternate members, and volunteer stewards who monitor town properties. The full and alternate members are elected to the local office. The members hold an annual town hall meeting for public input about the natural resources, including the forests. The volunteer stewards are the caretakers, maintaining trails for hiking, cross-country skiing, and mountain biking. The property does not contain a conservation easement but has an executory interest, meaning the town cannot sell the property, nor can it be deforested.

### 3.3. Case Study Summary

In summary, recreation is the most common use of the forests in the four case CF studies. In Birch Ridge and Page Pond CFs, snowmobiling is a widely practiced use of the forests in the winter. Two of the four cases (Page Pond and Nine Times) allow visitors to engage in consumptive services such as fishing and hunting. One of the four case studies (Río Hondo) provides non-timber forest products, such as small plants and herbs.

In addition, all four cases describe their CF as playing a crucial ecological role by offering regulating and supporting services for the community, such as clean water and air, carbon sequestration, and biodiversity. For example, interviewees at Birch Ridge indicated that the CF provides substantially healthier water quality and clarity of Merrymeeting Lake, which is the center of New Durham. In addition, members of Río Hondo recognize the value that the forest provides for the urban community, such as carbon sequestration and sheltering valuable wildlife.

The CFs involved in our case study analysis did not generate a large amount of revenue from timber or non-timber products, nor did they directly create a large number of local jobs. No direct paid jobs were reported for the four case studies that we visited. This may or may not be the case for other CFs in the Eastern U.S., and future research should examine this factor. Although Page Pond and Nine Times generate some income from timber harvests or hunting leases, timber income was not enough to sustain forest management on its own. All four case studies rely heavily on the support of their volunteers. From our literature review, we discovered this is not the case for many CFs outside of the U.S., but it occurs most commonly in developing countries where communities depend on the economic benefits that the CF provides [54,55]. Revenue generation was not the focus for many of these CFs, but rather their primary mission was to preserve land and provide cultural services. The CFs we visited all resemble a nature preserve, park, or other public green space more so than an active commercial timber-producing forest. The main difference between these CFs and traditional public parks was the ownership status and governance processes, particularly in terms of their public input and participation.

## 4. Discussion

CFs in the Eastern United States can be viewed as a hybrid forest governance approach that shares some ownership, governance, management, access, and benefits characteristics with both public and private forests. Consistent with the literature, we found that CFs in the Eastern U.S. display a diverse spectrum of forest ownership, governance, management authority, and uses [17,21,22,56].

### 4.1. Ownership

Individuals are more likely to protect forests or other natural resources when they have a sense of ownership and control [36,57,58]. CFs in the U.S. utilize several ownership models, most often varying by region. For example, we found four ownership categories represented in the Eastern U.S.: local governments, nonprofit organizations, tribal entities, and private for-profit corporations. Ownership by local governments appears to be the most common type in the East, including 50% in our search. CFs in the U.S. also may be owned by state and federal governmental agencies, though these ownerships are not known to exist in the Eastern U.S.

Town forests owned by local governments are most prevalent in the New England states, and many of these forests represent the key characteristics of CFs, including public participation and local community benefits. Town forests have highlighted the collaborative, community-based nature of conservation efforts; however, with limited focus on revenue generation and other economic-based benefits for the local communities [59]. Local governments throughout the Northeast also own CFs. CFs in the Eastern U.S. also may be under joint ownership, for example, held by local community nonprofit groups and land trusts or local government.

### 4.2. Governance

CFs in the U.S. span a wide range of governance and management frameworks [17,18]. In all but one of the cases in this study, we found that the landowner had the ultimate decision-making authority over the CF's management objectives, financials, and other duties of the property [36,52,60]. Geographic trends and histories can influence governance structures. For example, public input in decision-making processes is frequent in town forests in New England states, as demonstrated in our case studies, and consistent with the reports from the Community Forest Collaborative [61]. Town forests have been a part of New England's municipal history, and community members' role in decision-making processes is not new, dating to the early colonial settlements [61].

For three of our four case studies, Nine Times, Birch Ridge, and Page Pond, the landowners, being either a nonprofit organization (e.g., SELT and Naturaland Trust) or municipal government (e.g., Town of Meredith), are the final decision-makers. For Page Pond, large decisions, such as purchasing additional parcels, are voted on by the community members at town hall meetings. For minor questions, decisions are made by local elected officials of the Meredith Conservation Commission. For Birch Ridge, the CF steering committee, consisting of local citizens, incorporates public comments into the decision-making process. The committee gives their opinions and decision options to the landowner, SELT, who then has an ultimate say. The town of Mayagüez owns Río Hondo, but the nonprofit's Board of directors is the main decision-making body. Occasionally, a staff member from the municipality will join board meetings and contribute to board discussions and decisions.

### 4.3. Public Participation

Public participation and input in decision-making have been theorized to prevent conflict and promote the public's trust in government [62–64]. Public participation in forests may occur in decision-making and on-the-ground management. For instance, local community members partake in activities, such as trail-cleaning, non-timber and timber harvest, and property monitoring. From these CF case studies, we learned that public

participation in decision-making processes is not always a priority nor explicitly included in management decisions or processes. Some CFs seemed to mostly incorporate these elements to comply with CFP requirements that mandate incorporating some form of public participation.

CF public participation and input typically occurred through engagements in informational meetings and participation in site tours hosted by the CFs (e.g., Río Hondo). The election of local community members to decision-making bodies and management roles of the CF also was quite common (e.g., Page Pond). Less common for these CFs was widespread or deep participation by a large portion of local community members. Birch Ridge CF in New Hampshire was unique in its incorporation of in-depth public input and participation in forest decision-making.

The four case study CFs fall along different points of the International Association of Public Participation's (IAP2) public participation spectrum in decision-making processes [65]. The IAP2 encompasses five levels of public participation: inform, consult, involve, collaborate, and empower [65]. Some CFs minimally engage in public participation by informing the public via forest entrance signs and kiosks, hiking trail maps, or advertising community meetings or events (e.g., inform). On the other end of the spectrum, some CFs depend heavily on volunteers and local community members as the primary leaders and decision-makers (e.g., empower). Most CFs lie in the middle of the two extremes, many leaning towards IAP2's consult and involve levels of public participation. We found that our case study CFs and many of the CFs we identified in the Eastern U.S. involve the public by obtaining comments, including community members in direct decision-making conversations, or partnering with local groups and organizations to seek advice or resources.

### 4.4. Uses, Benefits, and Outcomes

In addition to a sense of ownership, people are more likely to protect and manage natural resources when they have secure access [36,57,58,66]. All four case studies allow some form of public access and mainly focus on social and ecological outcomes. The case study CFs did not generate much direct income and employment from timber or non-timber (e.g., hunting leases for Nine Times, NTFPs for Río Hondo) markets. The online search also indicated that profits from commodity production were seldom mentioned. Approximately 20% of the CF's website pages reference revenues from NTFPs, and more commonly, 64% of CF's websites mentioned timber production benefits.

The principal outcomes from our cases, and perhaps from many CFs to date, are the provision of ecosystem services across the four broad MEA classes of supporting, regulating, cultural, and provisioning services of forests [4]. CFs often are fomented by desires to set aside and protect pubic and common pool goods—regulating (e.g., water, wildlife habitat, and biodiversity, air quality, carbon storage), recreational or cultural (e.g., dispersed, hiking, water sports, climbing, birding), and more broadly supporting services (e.g., nutrient and soil formation and primary production) functions. While they also produce some modest income from provisioning timber harvests and non-timber goods or services, this use alone does not appear to be the primary cause for creating or keeping CFs. Most CFs primarily aim to preserve and protect the forest from future degradation or conversion, thereby protecting common-pool resources.

The cases studied also described social benefits from their CFs, including building a sense of community and place, engaging the community to become involved in natural resource management, and bringing local community members together. In addition, social outcomes from CF included other cultural uses such as environmental education and recreational purposes.

### 4.5. CF Challenges

From the four case studies, three of the CFs (Río Hondo, Nine Times, and Birch Ridge) struggled to obtain sufficient funding to either support full-time or part-time employees

and maintain features and functions of the CF, such as hiking and snowmobile trails, roads, signage, and parking lot installation. Financial resources are necessary to sustain a local and collaborative project for the long term [67].

Other challenges include recruiting and retaining community members to assist in public participation in decisions and management; balancing needs to attract visitors to promote community engagement with concerns for resource protection; and increasing the forest's visibility as a CF. For instance, Page Pond's grounds management crew consisted of retired and older individuals. The need for younger stewards to join the Page Pond efforts is essential for the CF's long-term viability. In addition, Río Hondo strived to increase the number of visitors to the forest. Other than local school and church groups that visited for site tours, the forest had one visitor a day on average at the time of data collection. Recruiting, motivating, and retaining volunteers and visitors is crucial for social and economic outcomes, especially for programs with limited resources [67].

*4.6. Study Limitations*

This exploratory research was intended to provide an initial descriptive introduction and analysis of CF efforts in the Eastern U.S. since related research and literature are extremely limited. The prospective benefits of CFs as a potentially promising model of forest ownership and management in the U.S. are large, but research information about this approach is scarce. Our lists of CFs are thorough, and the four cases selected are representative of the organizational and geographic range of CFs identified. Given the diversity of CFs nationwide and globally, our cases aim to provide a portrait of the range in types of CFs and their governance, public input processes, collaborative efforts, management plans, and recreation, commodity, and ecosystem benefits.

An apparent limitation is that we had only four detailed case studies and 18 persons interviewed. Nevertheless, the CFs visited, and the persons interviewed, were quite interesting and candid, and provided considerable insights into their experiences and prospects. A greater number of cases would be the next step to enhancing our knowledge about CFs, and we are continuing to examine this line of research in greater depth as well. The purposive cases examined here as drawn from our list of CFs, however, did serve as quite typical examples of relatively active New England CFs, where Town government owners and management have existed for decades, and of a nongovernment agency/land trust, which is increasingly common. The southern U.S. case with a land conservancy is a common approach, and the Puerto Rico case demonstrates how CFs can achieve forest protection and management as a promising alternative to development. The cases, merits, issues, and implications should be useful for guiding further research.

Last, the information obtained from online searches only represents the documentation made available online. Nevertheless, the case study information can give us an idea of which CFs are most active, what management and governing characteristics are most employed, and other general characteristics to help descriptive CFs in the Eastern U.S.

## 5. Conclusions

CFs reveal both direct and indirect benefits for local communities and the natural environment. In this study, we compiled a list of Eastern U.S. CFs through a systematic web-based search. We selected four CFs to conduct in-depth comparative case study research. Generally, we found that CFs benefits included better protected and more resilient biophysical resources, enhanced social capital, enabled local and indigenous knowledge in forest management and governance, and increased economic resources from timber and NTFP production. In addition, this study revealed that CFs in the Eastern U.S. have often been formed to protect forest land from development and protect associated common-pool resources. These efforts frequently bring together a group of diverse stakeholders, or at least a small number of very committed groups and individuals, to protect and manage the common-pool forest resources.

The initial efforts at CF establishment must be maintained for long-term viability and success. Maintaining and funding such efforts in the long run without significant revenues remains a challenge for most CFs. As CF initiatives increase in the Eastern U.S. and across the country, the information generated through this study can be used by CF practitioners, researchers, landowners, managers, and community members to further establish and maintain those efforts to increase ecosystem goods and services for local communities. In addition, a similar methodological framework could be adapted in other parts of the country to gain further information on the number of, types of, and characteristics of CFs in the United States.

**Author Contributions:** M.H., F.C., K.M. and G.F. contributed to writing the original draft, finalizing methodology protocols, and supervising the overall study. M.H., K.M. and X.H. conducted semi-structured interviews with community forest case study participants, and M.H. formally analyzed the qualitative data and curated it. All authors (M.H., F.C., K.M., G.F., X.H. and M.L.) contributed to online web search data collection, validation, and the analysis of community forests and characteristics descriptive statistics and contributed to the formation of the community forests in the Eastern United States list. All authors also contributed to reviewing and editing the manuscript. All authors have read and agreed to the published version of the manuscript.

**Funding:** Travel for this research was funded by the North Carolina State University Department of Forestry and Environmental Resources Zobel and Laarman Endowments.

**Institutional Review Board Statement:** The study was conducted in accordance with the Declaration of Helsinki, and approved by the Institutional Review Board of NORTH CAROLINA STATE UNIVERSITY (protocol code 16837, 1 April 2019).

**Informed Consent Statement:** Informed consent was obtained from all subjects involved in the study.

**Data Availability Statement:** The data are available on request from the corresponding author.

**Acknowledgments:** We thank the community forest case study participants for their time and participation in the interviews.

**Conflicts of Interest:** The authors declare no conflict of interest. The findings and conclusions in this publication are those of the authors and should not be construed to represent any official USDA or U.S. Government determination or policy.

## Appendix A  Keywords Search

**Table A1.** Keywords used in systematic search, using 13 mile woods community forest as an example.

| |
|---|
| "13 Mile Woods" + "Community forest" * |
| "13 Mile Woods" + "Area" |
| "13 Mile Woods" + "Association" |
| "13 Mile Woods" + "Mission" |
| "13 Mile Woods" + "Vision" |
| "13 Mile Woods" + "Purpose" |
| "13 Mile Woods" + "Manager" |
| "13 Mile Woods" + "Management" |
| "13 Mile Woods" + "Management plan" |
| "13 Mile Woods" + "Stewardship plan" |
| "13 Mile Woods" + "Annual report" |
| "13 Mile Woods" + "Owner" * |
| "13 Mile Woods" + "Strategic plan" |

**Table A1.** *Cont.*

| "13 Mile Woods" + "Committee" |
| --- |
| "13 Mile Woods" + "Board" |
| "13 Mile Woods" + "Members*" |
| "13 Mile Woods" + "Partners" * |
| "13 Mile Woods" + "Budget" |
| "13 Mile Woods" + "Income" |
| "13 Mile Woods" + "Revenue" |
| "13 Mile Woods" + "Recreation" |
| "13 Mile Woods" + "Education" |
| "13 Mile Woods" + "Timber" |
| "13 Mile Woods" + "Timber products" |
| "13 Mile Woods" + "Non-timber products" |
| "13 Mile Woods" + "Public access" |
| "13 Mile Woods" + "Users" |

* Represents keywords that have been truncated, meaning the web-based search include words with various word endings or phrases. For example, "community forest" could also search for phrases like "community forestry" or "community forests".

### Appendix B  Interview Questionnaire

**Introductory questions:**

- Can you discuss a bit about the history of the community forest?
- What role did you play in the establishment of the forest?
- What is your responsibility or position of the forest?
- **How has the forest changed over time?**

*Optional follow-up questions:*

- For example, in terms of ownership, size, organization, management, uses, productivity
- Are there any growth trends over time based on the organization's members, resources, and capacity? Can you give examples?

**(Q1) What activities occur on the forest?**

*Optional follow-up questions:*

- Does the CBF produce timber and wood products, the materials derived from a forest for commercial and personal use (but not limited to) pulpwood, firewood, and sawtimber?
- Does the CBF produce non-timber forest products (NTFPs), the materials derived from forestlands other than round wood or wood chips, including (but not limited to) fruits, nuts, honey, mushrooms, seeds? Is it a permit required to retrieve NTFPs from the CBF?
- Does the CBF provide for leisure activities such as hiking, walking, running, canoeing, kayaking, etc?
- What types of recreation activities are offered?
- What types of educational purposes to learn about forest management, conservation, wildlife, etc. for school groups and other community organizations are offered?

**(Q2) Who owns the forest?**

*Required follow up:*

- How has the ownership evolved since establishment of the forest?

*Optional follow-up questions:*

- Who has complete legal possession of the forestland or land rights?  Out of the following ownership categories:

○ Land trust or conservancy
○ Environmental or conservation nongovernment organization (NGO)
○ County, city, township, or other local government entity
○ Sole proprietorship
○ Joint Venture, General Partnership Limited Partnership
○ Limited Liability Company+ (LLC, LLP, LLLP)
○ Corporation
○ Trust
○ Tribal Reservation
○ Other

- Who has the right to obtain resources or products from the CBF (such as timber, fishing or diverting water)?
- Who has the right to regulate, change, and implement the CBF actions and its objectives?
- Who has the right to decide who has access rights and withdrawal rights?
- Who has the right to sell products or lease management on the CBF?

**(Q3) Who manages the forest?**
*Required follow up question:*

- How has management authority evolved since establishment of the CBF?
- Who performs the management work on the forest? (e.g., staff, contractors, volunteers, CBF Members?)

*Optional follow-up questions:*

- What is the manager's authority, title, and responsibility? What is the process of managing the forestland?
- Is this person (or organization) a paid member or volunteer?

**(Q4) How are decisions made?**
*Required follow-up question:*

- What individual or group has the final authority to make the decisions of the forest?

*Optional follow-up questions:*

- How are decisions made on the strategic level?
- How are decisions made on an operational/managerial level?
- What is the balance between Board and the CEO/EVP/professional staff authority in making decisions? In implementing decisions?
- How is the budget agreed upon by the board members and managers?

**(Q5) Can you discuss a little about the CBF's organizational structure?**
*Optional follow-up questions:*

- Does the CBF have a charter, bylaws, a strategic plan, operating policies, or management and operational regulations in place?
- Does the CBF have a formal organizational hierarchy and policies or practices manual?
- Is the CBF comprised of various businesses, nonprofits, or community organizations? Are there alliances involved in the management processes? Are the organizations a part of the private or public sector?
- Does the forest monitor the lands through rangers or other staff members?
- What are processes for resolving conflicts either between board members, staff, volunteers, or volunteers?

**(Q6) Who receives forest benefits?**
*Required follow-up question:*

- What are the ecological and human benefits, both locally and globally, provided by the forested land does the CBF recognize?

○ *e.g., Local: healthier water, cleaner air supply, food, wood and fiber, fuel, biodiversity habitats, education, recreation, spiritual and cultural opportunities (Millennium Ecosystem Assessment, 2005).*

    ○    *e.g., Global: climate regulation, flood regulation, disease regulation, and water purification (Millennium Ecosystem Assessment, 2005).*

*Optional follow-up questions:*

- Who is eligible to become a user?
- What does membership/usership require and/or cost?

**(Q7) Does the forest generate income? If yes, how so?**

*Optional follow-up questions:*

- (Such as sale of forest products, member dues, user fees, service sales, publications, conventions and meetings, advertising, gifts and contributions, interest and dividends from investments, etc.)
- Do you receive periodic or recurring state or federal grants for the CBF? How much per year?
- What funds were used to purchase/establish the forest property?
- Does the CBF provide any indirect impact? For example, indirect income is one which is earned by way of non-forest products and services, such as income through local gas stations, restaurants, and lodging.

**(Q8) What are the forest's expenses?**

*Optional follow-up questions:*

- What are the current assets— such as cash, receivables, prepaid expenses, land, property, forests, investments, buildings, equipment, and liabilities— such as lines of credit, accounts payable, deferred membership dues, etc.? If any, what are the unrestricted and restricted net assets?
- What funds are used to manage the CBF?
- If federal or state grant funds are received, do they require specific management actions, research, or other deliverables? Are they discretionary?
- If there is net profit or income in excess of costs, how are those funds used? Is any net profit distributed to community organizations or community members?

**Concluding questions:**

**(Q9)** Do you think the property should be owned and managed as a community forest, or might another public or private management system work better?

**(Q10)** What suggestions do you have for other communities that might be interested in community forestry? Under what conditions do you think a community works best together?

**(Q11)** What are effective public engagement processes or methods for developing a community forest?

**(Q12)** Who are the individuals or organizations that might oppose community forest development? What roles might they play? How can they be effectively engaged?

**(Q13)** What are the strengths of the community forest you are associated with?

**(Q14)** What are possible weaknesses or biggest challenges of the community forest you are associated with?

**(Q15)** What other opportunities could the forest you are associated with take advantage of?

**(Q16)** What threats are there to the community forest you are associated with?

**(Q17)** Do you have any final thoughts you would like to share about your community forest or your personal experience with the community forest?

## Appendix C

**Table A2.** List of eastern U.S. community forests identified through an online search.

| Community Forest | Location |
| --- | --- |
| 13 Mile Woods | Errol, NH, USA |
| Albany Town Forest | Albany, NH, USA |
| Albert Family Community Forest | East Nassau, New York, NY, USA |
| Barre Town Forest | Websterville, VT, USA |
| Bethel Community Forest | Bethel, ME, USA |
| Birch Ridge Community Forest | New Durham, NH, USA |
| Blackstrap Hill Community Forest and Preserve | Falmouth, ME, USA |
| Bobcat Woods Community Forest | Bolton, CT, USA |
| Browns Mill Community Forest | Atlanta, GA, USA |
| Brushwood Community Forest | West Fairlee, VT, USA |
| Buffam Brook Community Forest | Pelham, MA, USA |
| Canaan Community Forest | Canaan, VT, USA |
| Catamount Community Forest | Williston, VT, USA |
| Clarence Oak Savannah Community Forest | Clarence, New York, NY, USA |
| Cooley-Jericho Community Forest | Easton, NH, USA |
| Dead River Community Forest | Negaunee, MI, USA |
| Dorset Town Community Forest | Dorset, VT, USA |
| Downeast Lakes Community Forest | Grand Lake Stream, ME, USA |
| Falmouth Town Forest | Falmouth, ME, USA |
| Freedom Community Forest | Freedom, NH, USA |
| Gill Town Forest (also known as Blake Town Forest) | Gill, MA, USA |
| Glen Oakes Town Forest | Fremont, NH, USA |
| Gorham Town Forest | Gorham, NH, USA |
| Hadlock Community Forest | Falmouth, ME, USA |
| Hall Mountain | Franklin, NC, USA |
| Healing Harvest Forest Foundation | Copper Hill, VA, USA |
| Hidden Valley Nature Center | Jefferson, ME, USA |
| Hoke Community Forest | Raeford, NC, USA |
| Holland Community Forest (previously referred to as Sichol Family Property) | Holland, MA, USA |
| Jefferson Community Forest | Fairdale, KY, USA |
| Kirby-Ivers Town Forest | Pelham, NH, USA |
| Kuncanowet Community Forest | Dunbarton, NH, USA |
| Lakeview Federal Stewardship | Falmouth, ME, USA |
| Lime Lake Community Forest (Portman Nature Preserve) | Almena and Antwerp, MI, USA |

**Table A2.** *Cont.*

| Community Forest | Location |
|---|---|
| Lincoln Community Forest | Lincoln, WI, USA |
| Lowry Woods Community Forest | Madison, CT, USA |
| Machias Community Forest | Machias, ME, USA |
| McLaughlin Crossing Community Forest | Falmouth, ME, USA |
| Mendon Town Forest | Mendon, MA, USA |
| Menominee Forest | Neopit, WI, USA |
| Milan Community Forest | Milan, NH, USA |
| Niantic River Watershed (also referred to as East Lyme Land Trust Community Forest) | East Lyme and Waterford, CT, USA |
| Nine Times Community Forest | Pickens, SC, USA |
| North Falmouth Community Forest | Falmouth, ME, USA |
| North Pikes Creek Community Forest | Russell, WI, USA |
| Oak Ridge Town Forest | Fremont, NH, USA |
| Page Pond Community Forest | Meredith, NH, USA |
| Palmetto-Peartree Preserve | Columbia, NC, USA |
| Peabody Town Forest | Pelham, NH, USA |
| Perley Mills Community Forest | Denmark, ME, USA |
| Pilgrim Community Forest | Houghton, MI, USA |
| Plimpton Community Forest | Sturbridge, MA, USA |
| Poestenkill Community Forest | Poestenkill, NY, USA |
| Ragged Mountain Community Forest | Charlottesville, VI, USA |
| Raymond Community Forest | Raymond, ME, USA |
| Richmond Town Forest (Andrews Community Forest) | Richmond, VT, USA |
| Rines Community Forest | Cumberland, ME, USA |
| Rio Hondo Community Forest | Mayagüez, PR, USA |
| Sink Creek Community Forest | San Marcos, TX, USA |
| Springfield Bluffs | Springfield, WI, USA |
| Stemilt Squilchuck Community Forest | Richmond, VT, USA |
| Towar Woods | Meridian, MI, USA |
| Twin Bridges Community Forest | Otisfield, ME, USA |
| Tyler Forks Community Forest | Mellen, WI, USA |
| Virginia Stranahan Town Forest | Marshfield, VT, USA |
| Waitsfield Scrag Town Forest | Waitsfield, VT, USA |
| Weston Community Forest (previously called Weston Homestead Farm land) | Madison, ME, USA |
| Wildcat Falls | Haight, WI, USA |
| Woods Road Community Forest | Falmouth, ME, USA |
| Yellow Dog River Community Forest | Ispheming, MI, USA |

Note: This list of community forests was documented through an online search and has not been verified on the ground. Therefore, it is likely that the list includes some initiatives that are not community forests (e.g., defunct, or never started, or do not meet the characteristics of a community forest) and does not include some community forests that exist on the ground but less online presence.

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
