# Peer review of "Ownership, Governance, Uses, and Ecosystem Services of Community Forests in the Eastern United States"

_forests, doi:10.3390/f13101577_

Round 1

Reviewer 1 Report (Previous Reviewer 1)

I continue to believe that the manuscript is not suitable for such a respectable journal. There has been a considerable effort by the authors to improve some flows, e.g. the content of the text, but the overall merit of the survey cannot be improved.

There are issues, such as the structure of the methodology, conducting research on 18 participants/volunteers - who under no circumstances can be considered experts, the weaknesses of the research design, etc. that cannot be cured. As I proposed authors need to incorporate additional interviews, further information and robust statistical procedures.

Author Response

We thank you for your second review, perspective, and acknowledgment of the many revisions that we made. Unfortunately, due to the lack of funding and time, we do not have the ability to incorporate additional interviews.

For exploratory and case study research studies such as this one, a small sample size of participants is appropriate (See Yin, 2003, 2009, 2018; Clarke & Braun, 2013; Fugard & Potts, 2014; Guest, Bunce, & Johnson, 2006). With a triangulation of various methods (document review, systematic web-based searches, and semi-structured interviews), we feel that the methodological structure is robust for a qualitative study.

In addition, most researchers who closely study community-based research management and collaboration in natural resources agree that local communities that live near and on the forests, and manage the day-to-day activities, are the relevant experts in community forestry, and its management and establishment practices.

Thank you, again, and we are happy to answer any other questions that you may have.

Reviewer 2 Report (Previous Reviewer 2)

In my view, this manuscript has improved and will be a great contribution.

Author Response

Thank you for the positive feedback, and we appreciate your re-review and acceptance of our paper.

All the best,

The authors

Reviewer 3 Report (Previous Reviewer 3)

The authors have made some improvements over the initial version of the manuscript. The paper still needs an additional round of editing to clarify and strengthen the writing. In addition to the following specific comments, I recommend a thorough copy edit to fix numerous small errors in grammar, capitalization, use of commas, and the like.

Line 21: Change to clarify that you conducted interviews with CF stakeholders (not with CFs themselves)

Line 73: Spell out CFs before using the acronym

Lines 87-88: This needs more clarity; the writing is currently muddy. Change to something like “…agree that CFs are defined by the following characteristics: 1) local populations have a substantive role in decision-making; 2) local values are incorporated into management and governance; and 3) forests are managed to deliver material and nonmaterial benefits to local people”

Line 136: Change “from” to “about”

Line 237: Spell out CFP before using the acronym

Line 358: Clarify what you mean—it says “The majority of CFs mention…” but given that CFs don’t speak, what is the meaning here? Where is it mentioned what their purposes or intended uses are? Same in lines 360 and 362.

Lines 366-369 also need work. To say these are “premises” does not clarify what it meant here. Are these purposes or motivations for establishing a CF? If so, then the points should be changed to “to secure access to various goods and services”; “to allow for public participation…” “to protect the property from conversion to non-forest uses”

Author Response

Thank you for the overall positive response. We have made many grammatical revisions and have made the revisions as you suggested. Please see how we responded to your comments below in italics. We’ve attached both a track changes version and a clean version. For easier review, the line numbers we state below in our responses refer to the lines in the clean document version. Thanks, again, for your help in improving our paper for publication.

All the best,

The authors

Line 21: Change to clarify that you conducted interviews with CF stakeholders (not with CFs themselves)

Good point. See revision at line 22.

Line 73: Spell out CFs before using the acronym

Good catch. Done. See line 70.

Lines 87-88: This needs more clarity; the writing is currently muddy. Change to something like “…agree that CFs are defined by the following characteristics: 1) local populations have a substantive role in decision-making; 2) local values are incorporated into management and governance; and 3) forests are managed to deliver material and nonmaterial benefits to local people”

We agree that this sentence needed to be further clarified. Thank you for the wording suggestion.  We have made the changes at lines 85-88.

Line 136: Change “from” to “about”

Done. See line 145.

Line 237: Spell out CFP before using the acronym

Done. See line 224.

Line 358: Clarify what you mean—it says “The majority of CFs mention…” but given that CFs don’t speak, what is the meaning here? Where is it mentioned what their purposes or intended uses are? Same in lines 360 and 362.

Good point. Thank you for this comment. We’ve revised this at lines 236, 238, 241, and 605-606.

Lines 366-369 also need work. To say these are “premises” does not clarify what it meant here. Are these purposes or motivations for establishing a CF? If so, then the points should be changed to “to secure access to various goods and services”; “to allow for public participation…” “to protect the property from conversion to non-forest uses”

Done. Thank you for the suggestion here. We have revised these sentences. See lines 245-248.

Round 2

Reviewer 1 Report (Previous Reviewer 1)

It can be accepted in present form.

Author Response

Thank you again for your review and comments that helped improve our manuscript. We appreciate your time and thoroughness.

Best regards,

The authors

This manuscript is a resubmission of an earlier submission. The following is a list of the peer review reports and author responses from that submission.

Round 1

Reviewer 1 Report

I have the following major and minor comments for the Author of the manuscript.

General comment

Reading this very interesting title the reader expects to read a more exploratory manuscript than the description of some previous projects and the results from the interviews of 18 participants.

Specific comments

The abstract confuses the reader. In particular, the authors describe partially the methodology, then cite a Reference and then draw some conclusions, without understanding whether these arise from their research or the existing literature. In each case, Abstract needs to be rewritten.

Methodology: Authors describe a qualitative survey among 18 participants. They interviewed “volunteers, landowners, managers, community members “. It could be a reliable qualitative survey if focused on experts. However, a volunteer or a community member can not be considered as an expert. So, I think that the results lack sufficient reliability.

The questionnaire is in the Annex describe a SWOT analysis, as well. However, the corresponding results are not mentioned in the text.

Section 3.1: The results presented detailly in this section, focus on the description of previous projects. However, I think that it can not attract the research interest that a manuscript should have. They could be information provided by a Report, but not included in a research paper.

Conclusion

 Τhe essence of the conclusions is not clear.  I recommended you to re-write the discussion part.

A general comment

As authors mention and I understand this manuscript is “a precursor to more detailed work forthcoming”. I kindly recommend to authors to incorporate the additional results that will emerge from their survey, in order to compose a complete research article which will be of particular interest indeed.

Reviewer 2 Report

The article "An Exploration of Community Forests in the Eastern United 2 States" evaluated different types of community forests in eastern USA, through online web-based searches. The study is interesting but too local, which gives little relevance to the study in the international community.

Abstract

line 22. Delete reference [1].

line 22. I suggest deleting "Apparently.." from the abstract because you are talking about your result. It is or it is not...

line 23-24. I don't understand, what effort specifically?

Introduction

I suggest expanding the general information regarding community forests: the proportion of forest area at the national level represented, the types of forests that are included in such an administrative figure, etc.

Lines 37-43. And what is the perspective or interest at the governmental level (national, state, counties, etc.)?

Lines 43-46. The authors recognize differences between the administration and objectives of public and private community forests, but what particularities identify each of these? It is important since it is part of the objective of the study (ownership, governance).

I suggest deleting de sub-title. Integrate the objective into the main text.

Methods

This section needs to be reordered and clarify several methodological aspects.

Lines 67-76. Move this paragraph to the introduction.

Lines 77-80. Move to or merge this with subsection 2.2

Lines 82-83. What keywords did you use for initial online web-based searches? Readers should be able to understand the search steps to replicate this research.

Lines 87-88. Delete your future objectives or discuss that in the Discussion section. Focus on the current study.

Lines 111-112. Move this to results.

Line 114. Please, give the names of the four Case Studies.

Lines 116-120. The criteria for selecting the 4 case studies are not completely clear. "The four were chosen to be both representative and diverse..." in terms of what? biological diversity, forest types, culture, management schemes? Also, representative cases of what? at a geographic level (north, center, south), forest types? Please, be more specific.

I think lines 126-127 give more details about the selection criteria.

Line 135. How many representatives per each one?

Line 153. I suggest replacing "exploratory" with "descriptive".

Lines 152-159. I suggest move this paragraph to the Discussion section and developing the potential limitations of your study compared to other similar studies.

Results

Line 163. Please, define "the Eastern U.S. (east of the Great Plains)" or whatever you need to define in the Methods section. Here (Results) you should refer to "the area of interest", or "studied area".

Line 140. Delete reference if you refer to your own result.

Lines 187-193. It is not clear where these ecosystem services are recognized, in the objective of community forests, or the rationale for their creation? I think this is not clear in the methodology (data extraction); therefore the results are not clear.

Lines 195-197. Move to the Methods section.

In general, the results are descriptive but consistent with the methodology used. 

The discussion is scarce but consistent with the results obtained.

Unfortunately, this work does not address general concepts / global problems / or hypotheses that can be extrapolated to similar systems in other parts of the world.

Reviewer 3 Report

This manuscript presents the findings of exploratory research on the diversity, purposes, uses, and governance of community forests in the eastern U.S. It includes basic descriptive statistics on a database of U.S. community forests followed by findings from four in-depth case studies. Overall the manuscript is in fairly good shape, but to my mind it still needs some revisions to improve clarity and impact. The suggested changes are detailed below—to me, the most important is providing a stronger framing of community forestry in the U.S. prior to the methods section. As currently written, the framing is extremely brief and not adequate for what the authors are intending to accomplish.

Detailed feedback:

The Introduction section is quite brief; considering this is not followed by a literature review, I recommend expanding on the treatment of community forestry literature. There are many relevant sources already included in the cited literature; their findings and framings should be incorporated in more detail in order to frame the rest of the paper.

Lines 253-254: change “damages” to “damage”

Line 435 is not clear—the three parameters listed can be seen as defining criteria for community forests, but they are not “spectra.”

Line 440: change to “the most common use of the forests in the four case studies”

After reading the paper, I’m a bit puzzled as to the inclusion of the “ecosystem services” framing. The use of the four Millennium Ecosystem Assessment categories seems to be an afterthought and is not central to the purpose of the paper, which is to provide the results of an initial exploration of community forests in the eastern U.S.